# Wide Concentration Range of Tb^3+^ Doping Influence on Scintillation Properties of (Ce, Tb, Gd)_3_Ga_2_Al_3_O_12_ Crystals Grown by the Optical Floating Zone Method

**DOI:** 10.3390/ma15062044

**Published:** 2022-03-10

**Authors:** Tong Wu, Ling Wang, Yun Shi, Xintang Huang, Qian Zhang, Yifei Xiong, Hui Wang, Jinghong Fang, Jinqi Ni, Huan He, Chaoyue Wang, Zhenzhen Zhou, Qian Liu, Qin Li, Jianding Yu, Oleg Shichalin, Evgeniy Papynov

**Affiliations:** 1State Key Laboratory of High Performance Ceramics and Superfine Microstructure, Shanghai Institute of Ceramics, Chinese Academy of Sciences, Shanghai 200050, China; wutong1@mails.ccnu.edu.cn (T.W.); wll1327201@outlook.com (L.W.); zq1421849356@163.com (Q.Z.); xiongyifei1996@163.com (Y.X.); wanghui@mail.sic.ac.cn (H.W.); fangjinghong@mail.sic.ac.cn (J.F.); nijinqi@mail.sic.ac.cn (J.N.); ch.hh@mail.sic.ac.cn (H.H.); cywang@mail.sic.ac.cn (C.W.); zhouzhenzhen@mail.sic.ac.cn (Z.Z.); qianliu@mail.sic.ac.cn (Q.L.); liqin@mail.sic.ac.cn (Q.L.); yujianding@mail.sic.ac.cn (J.Y.); 2College of Physical Science and Technology, Central China Normal University, Wuhan 430079, China; 3Center of Materials Science and Optoelectronics Engineering, University of Chinese Academy of Sciences, Beijing 100049, China; 4School of Materials Science and Engineering, Zhengzhou University, Zhengzhou 450001, China; 5College of Material Science and Engineering, Nanjing Tech University, Nanjing 211816, China; 6Laboratory of Nuclear Technology, Institute of High Technologies and Advanced Materials, Far Eastern Federal University, 690091 Vladivostok, Russia; oleg_shich@mail.ru (O.S.); papynov@mail.ru (E.P.)

**Keywords:** optical floating zone method, Ce^3+^ and Tb^3+^, scintillator, GGAG:Ce, Tb

## Abstract

To obtain a deeper understand of the energy transfer mechanism between Ce^3+^ and Tb^3+^ ions in the aluminum garnet hosts, (Ce, Tb, Gd)_3_Ga_2_Al_3_O_12_ (GGAG:Ce, Tb) single crystals grown by the optical floating zone (OFZ) method were investigated systematically in a wide range of Tb^3+^ doping concentration (1–66 at.%). Among those, crystal with 7 at.% Tb reached a single garnet phase while the crystals with other Tb^3+^ concentrations are mixed phases of garnet and perovskite. Obvious Ce and Ga loss can be observed by an energy dispersive X-ray spectroscope (EDS) technology. The absorption bands belonging to both Ce^3+^ and Tb^3+^ ions can be observed in all crystals. Photoluminescence (PL) spectra show the presence of an efficient energy transfer from the Tb^3+^ to Ce^3+^ and the gradually quenching effect with increasing of Tb^3+^ concentration. GGAG: 1% Ce^3+^, 7% Tb^3+^ crystal was found to possess the highest PL intensity under excitation of 450 nm. The maximum light yield (LY) reaches 18,941 pho/MeV. The improved luminescent and scintillation characteristics indicate that the cation engineering of Tb^3+^ can optimize the photoconversion performance of GGAG:Ce.

## 1. Introduction

The scintillator crystals with garnet structure have been proved to possess good chemical stability, high lattice isomorphic capacity and effective conversion of high energy ionizing radiation to visible luminescence when doped with luminescent rare earth ions in the host [1,2]. Those various garnet structure compounds, when coupling with photodetectors, are widely used for applications in industry inspection, nuclear medicine, and high energy physics [3,4]. Ce^3+^ doped Gd_3_(Ga, Al)_5_O_12_ (GGAG) is considered a promising new generation of scintillators in view of its allowed electric dipole 5d–4f transition, high theoretical light yield (LY), i.e., 73,500 pho/MeV [5], high density (6.63 g/cm^3^) and effective atomic number (Z_eff_ = 54), as well as hard radiation hardness performance [6].

Till now, the successful growth of GGAG:Ce crystals by melt method have been widely reported [7,8,9,10]. However, instability of the molten zone due to the incongruent melting trend in low Ga/Al ratio GGAG compositions remarkably affect the optical quality and performance by introducing cracks and composition un-uniformity [10,11,12]. Possible solutions to improve the molten zone stability include the introduction of a composition engineering in the host or modification of growth technology, such as using the traveling solvent floating zone (TSFZ) method. In the last decade, composition engineering strategy has demonstrated its positive effects in scintillation performance improvement and the crystal structure stabilization through changing the crystal field strength and the energy level position of the luminescence activators in the bandgap [8,13,14,15,16]. To stabilize the garnet crystal lattice, the cations at the dodecahedral can be replaced by the smaller Ln^3+^ ions (Ln = Y, Lu, Tb) [8,14,17], the octahedral and tetrahedral lattice sites can be substituted by the larger M^3+^ (M = Sc, Ga) [17,18,19]. Furthermore, doping rare-earth ions which is smaller than Gd^3+^ at the dodecahedral site was proved to be able to facilitate the garnet phase formation (Ln_3_Al_5_O_12_, LnAG) [20,21,22]. Among them, Tb^3+^ is regarded as a suitable co-activator to achieve the energy transfer since its absorption levels located between absorption of the host lattice and Ce^3+^ emission, and its 5d_4_–7f_6_ transition overlaps the 4f–5d_1_ absorption transition energy of Ce^3+^ ion [23,24].

Ogiegło, J.M. et al. [24] reported that efficient energy transfer from Tb^3+^ to Ce^3+^ where fast diffusion over the Tb^3+^ sublattice is followed by efficient emission from Ce^3+^ in the co-doped system LuAG:Ce, Tb. It was also found that substitution of the smaller Lu^3+^ ion by the larger Tb^3+^ ion would shift the lowest 5d state of Ce^3+^ to lower energies, and therefore the red shift of the Ce emission could be observed, even below the 5d_4_ state of Tb^3+^. Subsequently, Xin Teng et al. [25] investigated the Tb^3+^ concentration effect (between 0.07 at.% and 0.15 at.%) in Ce:(Gd, Tb)AG phosphors and successively shorter lifetime was found with the Tb^3+^ concentration increasing. Recently, Ce^3+^, Tb^3+^ co- doped Gd_3_(Ga, Al)_5_O_12_ garnet transparent ceramics, single crystalline films (SCF) and single crystal (SC) have been extensively studied [26,27,28,29]. Under α-particles excitation, the Tb_1.5_Gd_1.5_ Al_3_Ga_2_O_12_:Ce SCF grown from the PbO/BaO-based flux, LY is comparable with that of high-quality Gd_3_Al_2.5_Ga_2.5_O_12_:Ce single crystal (SC) scintillator, but its scintillation decay time is slow, which may be related to the contamination by lead ions from the PbO/BaO-based flux [30]. The growth, luminescent properties and energy transfer processes in Ce^3+^ doped Tb_3−x_Gd_x_Al_5-y_Ga_y_O_12_ (x = 0–1.5 and y = 2–2.5) SCs grown by μ-PD method [26] were also reported, the inhomogeneity of the dopant distribution in the SCs negatively influence their LY and decay time characteristics [8,14].

Hence, it is necessary to investigate Tb^3+^ concentration effect in GGAG:Ce crystals by OFZ method, which can allow us to obtain the crystals in an efficient way for the new material exploration. In this work, 1 at.% Ce^3+^ and x Tb^3+^ co-doped Gd_3_Ga_2_Al_3_O_12_ (GGAG:Ce, xTb) garnet crystals were grown by OFZ method, where x = 1 at.%, 7 at.%, 15 at.%, 33 at.%, 50 at.% and 66 at.%, respectively. The decomposition of Ga_2_O_3_ (Ga_2_O_3_→Ga_2_O + O_2_) and oxidation of Ce^3+^ and Tb^3+^ at high temperature were inhibited by N_2_ + 30% CO_2_ (CO_2_→O_2_ + CO) atmosphere. The chemical composition, luminescence and scintillation properties were also characterized.

## 2. Experimental Procedures

### 2.1. Crystal Growth

High pure CeO_2_, Gd_2_O_3_, Ga_2_O_3_, Al_2_O_3_ and Tb_4_O_7_ commercial powders (>4N) were used as starting material. They were mixed according to 6 kind of chemical stoichiometric of (Ce_0.01_Tb_x_Gd_0.99-x_)_3_Ga_2_Al_3_O_12_ (abbreviated as GGAG:Ce, xTb) (x = 1%, 7%, 15%, 33%, 50% and 66%, respectively), i.e., the seven compounds can be represented by Ce_0.03_Tb_0.03_Gd_2.94_Ga_2_Al_3_O_12_, Ce_0.03_Tb_0.21_Gd_2.76_Ga_2_Al_3_O_12_, Ce_0.03_Tb_0.45_Gd_2.52_Ga_2_Al_3_O_12_, Ce_0.03_Tb_0.99_Gd_1.98_Ga_2_Al_3_O_12_, Ce_0.03_Tb_1.485_Gd_1.485_Ga_2_Al_3_O_12_, Ce_0.03_Tb_1.98_Gd_0.99_Ga_2_Al_3_O_12_, respectively. Then they were sintered at 1350 °C for 8 h in air. After that, the powders were ball milled again and then rubber-pressed under a hydrostatic pressure of 70 MPa to form green ceramic rods which were sintered in success at 1600 °C for 8 h in tube furnace with fluent O_2_ atmosphere. The crystal growth apparatus used is an optical floating zone furnace (FZ-T1000H CSC Cop., Tokyo, Japan), in which halogen lamps of 4 × 1 kW were set as heating source focused by four polished elliptical mirrors. The rotate rate of a ceramic rod and seed crystal was tuned in the 15–20 rpm range to realize full stirring of the molten zone. Subsequently, the molten zone moved continuously at rate of 3–5 mm/h and N_2_ atmosphere (containing 30% CO_2_) with auxiliary pressure of 0.3 MPa were slowly introduced into the quartz tube and maintained during crystal growth. The growth rate was kept at 3 mm/h during the whole process. The obtained crystals were cut into wafers perpendicularly to the growth direction for measurement of scintillation properties. Crystals were double face polished to 2 mm thickness for the evaluation of their optical absorptance.

### 2.2. Characterization

All measurements were carried out at room temperature. The crystals were crushed and ground into powders for powder X-ray diffraction (PXRD) measurement in the 2θ range of 10–90° using the Rigaku Ultima IV diffractometer (Cu-Kα, 40 kV, 40 mA, Rigaku Ultima IV, Tokyo, Japan); scanning speed was 5 °/min, the baseline of the obtained data was subtracted by Jade6 software. The X-ray photoelectron spectroscopy (XPS) measurement was conducted in a XPS spectrometer (ESCALAB 250, ThermoFisher, London, UK) using the GGAG:Ce, 7% Tb and GGAG:Ce, 55% Tb crystals, high pure (>4N) commercial Tb_4_O_7_ powders were used as reference sample. The SEM-EDS mapping images were characterized by a high-resolution SEM (S-3400 N, Hitachi, Tokyo, Japan) equipped with an energy dispersive X-ray spectroscope (EDS).

Absorption spectra in UV-Vis-NIR regions were recorded by the Varian Cary 5000 spectrometer. The photoluminescence (PL) and excitation (PLE) spectra were characterized on a Horiba FluoroMax-4 (Edinburgh, UK) fluorescence spectrometer. The PL quantum yield (QY) were measured by MCPD-7000 multichannel spectrometer (Otsuka Electronics Company, Osaka, Japan) under the excitation of 450 nm wavelength light.

Stead-state radioluminescence spectra of the samples were measured by the self-assembled X-ray fluorescence spectrometer (X-ray tube: 70 kV, 1.5 mA). LY values of the seven compound crystals were obtained from the pulse height spectra measured using ^137^Cs as the radiation source of γ-ray with a Hamamatsu R878 photomultiplier (1 kV) (Hamamatsu Photonics K. K, Hamamatsu, Japan).

## 3. Results and Discussions

The crystals were grown under the same crystal growth parameters to finally obtain the crystal rods with a length of 40–50 mm and diameter of about 6 mm. The color of crystals changed with the Tb^3+^ concentrations, as shown in Figure 1. The crystal with lower Tb^3+^ concentration is a pale-yellow color, whereas the samples with 66% Tb^3+^ concentration are yellow-brown. All crystals rods have a regular shape, indicating a stable crystal growth process.

Powder X-ray diffraction was performed to identify phases for the pre-sintered raw material powders and the as grown crystals. In Figure 2, the XRD analysis reveals that the ceramic powders, after sintering at 1350 °C, contain detectable secondary perovskite phases (see Figure 2a), while the patterns from crystal powders show that there is single garnet phase in GGAG:Ce, 7 at.% crystal, but in the rest Tb^3+^ doped components (x = 1%, 15%, 33%, 50% and 66%), detectable perovskite (GdAlO_3_, GAP) phase could be observed, see Figure 2b). It might be related to the Tb^3+^ doping saturation effect owing to lattice mismatch, as well as the decreasing of crystal field symmetry after more replacement of Tb^3+^ to Gd^3+^ ions in a dodecahedral position. The stability of GGAG lattice structure can be achieved by partially substituting Gd^3+^ with smaller rare earth ions, while a radius of Ce^3+^ ions is larger [31]. While with the increasing of Tb^3+^ concentration, as shown in Figure 2c, the prominent peak slightly shifts to a higher angle, which indicates the shrink of unit cells according to the Bragg equation (2dsinθ = nλ), and the lattice parameters measured by refinement decrease from 12.2286 Å to 12.1744 Å for the substitution of Gd^3+^ (1.05 Å) by the smaller Tb^3+^ (1.04 Å) ions in garnet structure [32]. We also analyzed the percentage of second GAP phase of each component, and the results were summarized in Figure 2d. It can be seen that most of the GGAG:Ce, Tb crystals contains second perovskite phase except 7% Tb^3+^ case. It was proposed that GdAG will decompose at 1500 °C to GAP and Al_2_O_3_ [32,33], and at least 50% Tb^3+^ substitution to Gd^3+^ is needed to stabilize the GdAG structure, while Ga^3+^ substitution to Al^3+^ is helpful to expend and stabilize the lattice. However, in this study 7% Tb^3+^ can obtain single garnet phase, and the Laue photograph demonstrated the formation of integral crystal (Figure 2e). Besides, it seems that there is a random variation in the dependence of the second GAP phase percentage and Tb concentration in Figure 2d. Figure 2e presented the Laue photograph of doped GGAG:Ce, 7% Tb crystal. The Laue diffraction spots (white dot) were clear with good symmetry, which match well with that of the standard pattern of Gd_3_Ga_2_Al_3_O_12_ (GGAG) (100), indicating a high crystallinity of GGAG:Ce, 7% Tb crystal.

The results in previous published work [33,34] show that GdAG will decompose at 1500 °C to GAP and Al_2_O_3_, and at least 50% Tb substitution to Gd is needed to stabilize the GdAG structure, while Ga^3+^ substitution to Al^3+^ is helpful to expend and stabilize the lattice. However, in our case, 7% Tb^3+^ can obtain a single garnet phase. We suspect the formation of a second phase might be due to (1) the composition deviation origin from the evaporation of Ga, Ce during high temperature crystal growth process; (2) The formation of Tb^4+^ origin from the N_2_ atmosphere (containing 30% CO_2_), oxygen produced by CO_2_ decomposition at high temperature may promote the oxidation of Tb^3+^ (1.04 Å, 8 coordination number) to Tb^4+^ (0.88 Å, 8 coordination number) [35], resulting in a sharp reduction of lattice constant and further increasing the degree of lattice distortion. Serious lattice distortion occurs when the Tb^3+^ doping content is too high or too low, and that is probably the formation of a second perovskite phase; this will be further investigated by the following XPS and EDS measurement.

Figure 3 shows the XPS spectra which explore the valence state of Tb^3+^ in the as grown GGAG:Ce, 7% Tb and GGAG:Ce, 50% Tb crystals using commercial Tb_4_O_7_ powder as reference sample. Due to the detect limitation of the instrument, the valence signal of Ce^3+^ cannot be well separated since it’s doping concentration (1 at.%) is low. The fitting results of valence state energy levels of corresponding orbitals are summarized in Table 1. According to the study of Nagpure, I.M. et al. [36], we carried out the peak fitting of 3d_3/2_ and 3d_5/2_ orbitals of Tb elements. As shown in Figure 3, the spectrum line of 3d_5/2_ and 3d_3/2_ of Tb^3+^ can be divided into two sub-bands, respectively, indicating that Tb^3+^ and Tb^4+^ co-exist in the compounds. In addition, the valence ratio of Tb elements is listed in Table 1, i.e., Tb^3+^ ratio (defined as Tb^3+^/(Tb^3+^+Tb^4+^)). The ratio of Tb^3+^ in GGAG:Ce, 50% Tb crystal is similar to that of in Tb_4_O_7_ commercial powders, which is much higher than that in a GGAG:Ce, 7% Tb single crystal.

The SEM-EDS mapping images for Ce, Tb element of the GGAG:Ce, 7% Tb and GGAG:Ce, 55% Tb crystals were presented in Figure 4a,c and Figure 4b,d, respectively. All of the 6 compounds were systematically investigated, and homogeneous element distribution can be found. Moreover, the deviation to nominal stochiometric concentration of Ce, Tb, Gd, Ga, Al elements were listed in Table 2, and an obvious loss of Ce and Ga element was revealed, which might be due to an evaporation effect during the high temperature crystal growth process. The origin of second phases observed in Figure 2 can thus be mainly ascribed to the Ce and Ga evaporation introduced composition deviation, as well as the formation and existence of Tb^4+^ valence state in those crystals.

The absorption spectra in the 200–800 nm range of the GGAG:Ce, xTb crystals are shown in Figure 5. All spectra show characteristic broad absorption in 425–475 nm range, which are ascribed to the allowed 4f–5d_1_ transitions of the Ce^3+^ ions in the garnet host, and the wide flat absorption phenomenon might be due to the energy level overlapping effect and/or absorption saturation [10]. It is worth noting that the 4f→5d_2_ (E2) transition of Ce^3+^ ions located approximately at 325–350 nm distorted by last Tb^3+^ absorption band when Tb^3+^ (≥7%) concentration, and 4f–5d_2_ absorption bands of Ce^3+^ ions are only observed in GGAG:Ce,1% Tb. The high saturation absorption in UV below 360 nm is unlikely to be due to the Ce^4+^ charge transfer band, since we used a N_2_ + CO_2_ gas mixture as the crystal growth atmosphere. Hence, the disappeared 4f–5d_2_ absorption bands of Ce^3+^ ions may result from the characteristic strong absorption of Tb^3+^ ions at the higher energy region that overlaps the nearby absorption peaks of Ce^3+^ and Gd^3+^ ions.

To further investigate the energy transfer mechanism and luminescence of Ce^3+^, Tb^3+^ and Gd^3+^ in GGAG garnet host, the photoluminescence (PL) and excitation (PLE) spectra of as grown GGAG:Ce, xTb crystal were measured at room temperature and shown in Figure 6. The excitation spectrum of Ce^3+^ emission at 530 nm is composed of three sets of broad bands, which belong to the characteristic excitation transition of Ce^3+^ and Tb^3+^ ions, respectively. The band peaking at around 340 nm and 450 nm were commonly ascribed to 4f–5d_2_ and 4f–5d_1_ transition of the Ce^3+^, respectively [37,38]. The sharp lines observed at 302–313 nm are related to the characterized 8S_7/2_–6I_j_, 6P_j_ excitation transition of Gd^3+^ ion [39]. In addition, it can be seen that the weak excitation bands centered at 319 nm and 372 nm related with due to (4f–5d_1_) Tb^3+^ and (7f_6_–5d_3_) Tb^3+^ transitions [26,40], respectively. In the nominal concentration range of 1–66 at.% Tb^3+^, GGAG:Ce, 7% Tb crystal was found to have the strongest excitation peak centered at 275 nm which might be corresponding to the Tb^3+^ 4f-5d_2_ transition. However, it overlaps partially with the f-f transition of Gd^3+^, i.e., ^8^S–^6^I. The presence of Tb^3+^ and Gd^3+^ related excitation bands monitored at Ce^3+^ emission indicates an occurrence of Gd^3+^/Tb^3+^→Ce^3+^ non-radiative energy transfer in the GGAG garnet host [28,29,41].

An obvious strong emission located at 490 nm, 543 nm, 591 nm, and 625 nm corresponding to the 5d_4_–7f_6_, 5d_4_–7f_5_, 5d_4_–7f_4_, and 5d_4_–7f_3_ transitions of Tb^3+^ can be observed [25] in Figure 6b, when excited by the characteristic Gd^3+^ and Tb^3+^ related excitation wavelength (275 nm). Besides, a broad emission in the 490–700 nm range can be observed, which is related to the broad 5d_1_–4f transition of Ce^3+^. It is reasonable to assume that the presence of effective energy transfer from the host lattice to the Ce^3+^ activator ion. In the case of GGAG:Ce,1% Tb crystals, an emission band located at 383 nm can be observed which could be attributed to GAP: Ce, a second phase in GGAG:Ce, Tb crystal related emission [31]. In Figure 6c, the PL spectra were recorded for Tb^3+^ excitation at 320 nm. It is interesting to note that the characteristic emission of Tb^3+^ is broadened and intensity decreased obviously by the board emission band of Ce^3+^, since its 5d_4_–7f_6_ emission transition energy is basically resonant with the 4f–5d_1_ absorption transition of Ce^3+^ ion [23,24].

In Figure 6d, the photoluminescence spectra are shown under excited at 450 nm for GGAG:Ce, xTb crystals. The emission bands centered at 540 nm belongs to the 5d_1_–4f transition of Ce^3+^ in the garnet phase. In addition, it was found that the PL intensity under different excitation wavelength significantly decreased with increasing of Tb^3+^ concentration. The highest integral luminescence intensity in the 450–700 nm range is observed in lower Tb^3+^ concentration (see Figure 7), in accordance with the Tb^3+^ and Gd^3+^ cations in GAG can sensitizer Ce^3+^ to enhance the Ce^3+^ luminescence intensity. The Tb^3+^ substitution caused significant changes in the position of the Ce^3+^ emission bands related to garnet phase, as seen from the normalized photoluminescence (PL) (under the 450 nm excitation) spectra (in Figure 6e), the maximum of emission peak at around 540 nm shifts to lower energy (blue wavelength shift) with the increasing of Tb^3+^ content. Observed blue shift is due to weakening of the crystal field of the GGAG:Ce, Tb crystal. The 5d_1_ energy level of Ce^3+^ moved upward while the 5d_2_ level moved in the opposite direction, with replacing Tb^3+^ for Gd^3+^ (decreasing ionic radius) in the octahedral sites in the garnet structure [42,43].

It has become the norm to list measurements of QY when discussing luminescent materials [44]. The measurement of PL external QY is generally divided into three steps: Firstly, the 450 nm excitation light source was directly irradiated into the integrating sphere without any subjects to measure the total excitation spectrum; secondly, the crystal was placed in the integrating sphere, and the excitation light source irradiated while avoiding the sample to obtain the emission spectrum after secondary absorption; finally, the excitation light source was directly irradiated on the sample to measure the sum of the emission light of the sample origin from both primary and secondary absorption; i.e., the total emission spectrum of the sample. Therefore, the specific calculation formula of external quantum light yield is according to:(1)$$\eta =\frac{\int_{\lambda_{1}}^{\lambda_{2}}P_{em}\left(\lambda \right)\lambda d\lambda }{\int_{\lambda_{1}}^{\lambda_{2}}P_{abs}\left(\lambda \right)\lambda d\lambda }=\frac{E-C}{\left(A-D\right)-\left(A-B\right)}=\frac{E-C}{B-D}$$
where *A* is integral area of total excitation spectrum, *B* and *D* is integral area of the excitation spectrum after secondary absorption in second and third step, respectively; *C* and *E* is integral area of the emission spectrum in second and third step, respectively.

The calculated PL external QY of the as-grown GGAG:Ce, xTb (x = 0–66 at.%) crystals were shown in Figure 8. The GGAG:Ce, xTb crystals performed optimum QY (430–700 nm) when x = 15%, and then QY value decreased from 48.6% to 33% with the increase of Tb^3+^ concentration, which might be ascribed to the concentration quench effect. 

The scintillation properties of GGAG:Ce^3+^, xTb^3+^ (x = 1–66 at.%) crystals were also investigated. The XEL spectra of the crystals were presented in Figure 9. Besides the broad band located at 500–700 nm range which belongs to the internal 5d_1_–4f (2F_5/2_, 2F_7/2_) emission transition of Ce^3+^, it’s interesting to note that the sharp lines peaking at around 490 nm, 543 nm and 591 nm, they were considered to be related to the 5d_4_–7f_6_, 5d_4_–7f_5_, 5d_4_–7f_4_, and 5d_4_–7f_3_ transitions of Tb^3+^, respectively. However, the emission peak of Tb^3+^ decreased apparently in the heavily Tb^3+^doped crystals, which are consistent with the PL intensity changing trend shown in Figure 7.

The different intensity dependence on Tb^3+^ concentrations between PL and RL spectra may be due to different luminescence mechanisms, as well as different defect concentrations between crystals. For scintillation luminescence under high energy X-rays, there includes the process of the energy transfer and charge trapping in the transfer stage, which is complex, radiative electron-hole recombination at the luminescent center emission center Ce^3+^ (in the garnet phase) is the main contribution. The re-trap of charge carriers at shallow traps (i.e., energy levels introduced by material point defects, flaws, the second phase, surfaces and interfaces) may result in a considerable loss or delay [15]. GAP:Ce related emission located at high energy region can be observed in doped GGAG:Ce, 1% Tb crystal under 275 nm and 312 nm excitation, respectively, as shown in Figure 4c and Figure 6b, which may be considered as another origin to further enhance the luminescence intensity [45]. However, for photoluminescence, only the 5d–4f level transition of Ce^3+^ is monitored under the 450 nm excitation.

Pulse height spectra of GGAG:Ce^3+^, xTb^3+^ (x = 1–66%) crystals excited by ^137^Cs were obtained as shown in Figure 10. All the crystals were placed in darkness for 12 h before measurement to reduce phosphorescence. The relative scintillation LY value of the as grown crystals was calculated by comparing with reference GGAG:Ce crystal (absolute LY 58,000 pho/MeV, size: 27 × 15 × 2.37 mm^3^) with corresponding pulse height peak at 481.6, FWHM 33.60, E.R 6.98% and presented in Figure 11.

3–4 samples of each component were selected for a light yield test to ensure the reproducibility, as shown in Appendix A. The light yield value of all samples fluctuated in a fine range, indicating the uniform element distribution in the crystals. Besides, it is noted that the decrease of LY values with Tb^3+^ concentration is in a similar trend with that of QY value (see Figure 8). The light yield value in high Tb^3+^ concentrations decreased remarkably which might be due to the concentration quench effect. The GGAG:Ce, xTb crystals perform optimum LY when x = 7%, reaching to 18,841 pho/MeV, whereas in XEL the peak intensity is observed when x = 15%. This difference indicates that the GGAG:Ce, 7% Tb may probably have a lower defect concentration considering the 0.75 μs time gate [46] were used in scintillation LY measurement while XEL spectra give overall stead state radioluminescence intensity.

## 4. Conclusions

(Ce_0.01_Tb_x_Gd_1-x_)_3_Ga_2_Al_3_O_12_ (GGAG:Ce^3+^, xTb^3+^) (x = 0, 1%, 7%, 15%, 33%, 50% and 66%, respectively) scintillator crystals with uniform diameter were successfully grown by the OFZ method, which were rarely reported before. The XRD patterns show that integral cubic garnet structure can be obtained in GGAG: 1% Ce^3+^, 7% Tb^3+^ crystal. All crystals show characteristic broad absorption in 425–475 nm range, which originated from the allowed 4f–5d_1_ transition of Ce^3+^. The presence of Tb^3+^ excitation bands at lower Tb^3+^ concentration and the disappearance of Tb^3+^ emission lines at higher Tb^3+^ concentration under monitoring the Ce^3+^ emission demonstrate the occurrence of Tb^3+^→Ce^3+^ energy transfer. Obvious strong emission band located at 490 nm, 543 nm, 591 nm, and 625 nm corresponding to the 5d_4_–7f_6_, 5d_4_–7f_5_, 5d_4_–7f_4_, and 5d_4_–7f_3_ transitions of Tb^3+^, when excited by the characteristic Gd^3+^ and Tb^3+^ related absorption wavelength (320 nm and 275 nm, respectively). In the nominal Tb^3+^ concentration range of 1–66 at.%, the optimal content of Tb^3+^ is 7 at.%, which was found to have the most intense PL intensity and the comparatively highest LY value (18,941 pho/MeV). Therefore, with high emission intensity and good crystallinity, as well as high scintillation efficiency, the GGAG:Ce, Tb crystals are expected to be potential candidate for the application of high energy detection or white LED lighting.

## Figures and Tables

**Figure 1 materials-15-02044-f001:**
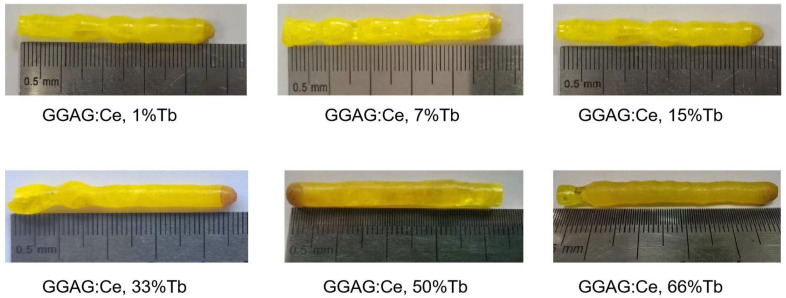
Photographs of the 1 at.% Ce doped (Tb_x_Gd_1-x_)_3_Ga_2_Al_3_O_12_ (GGAG:Ce, xTb) (x = 0–66 at.%) crystals grown by the optical floating zone method.

**Figure 2 materials-15-02044-f002:**
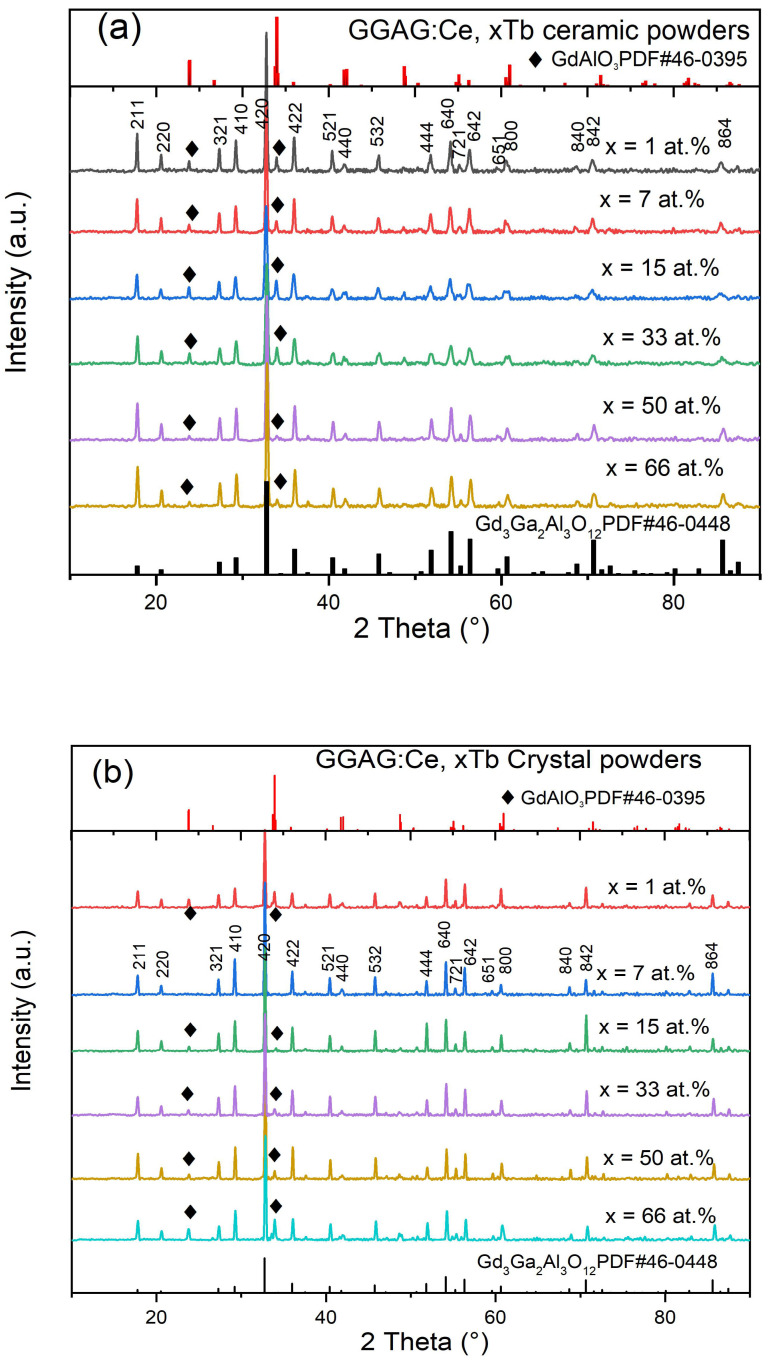
Powder XRD pattern of (**a**) the powders sintered at 1350 °C for 8 h in air, (**b**) GGAG:Ce, xTb crystals, (**c**) expanded view of diffraction peaks between 32° and 38°; (**d**) calculated percentage of perovskite phase; (**e**) Laue photograph of GGAG:Ce, 7% Tb^3+^ crystal.

**Figure 3 materials-15-02044-f003:**
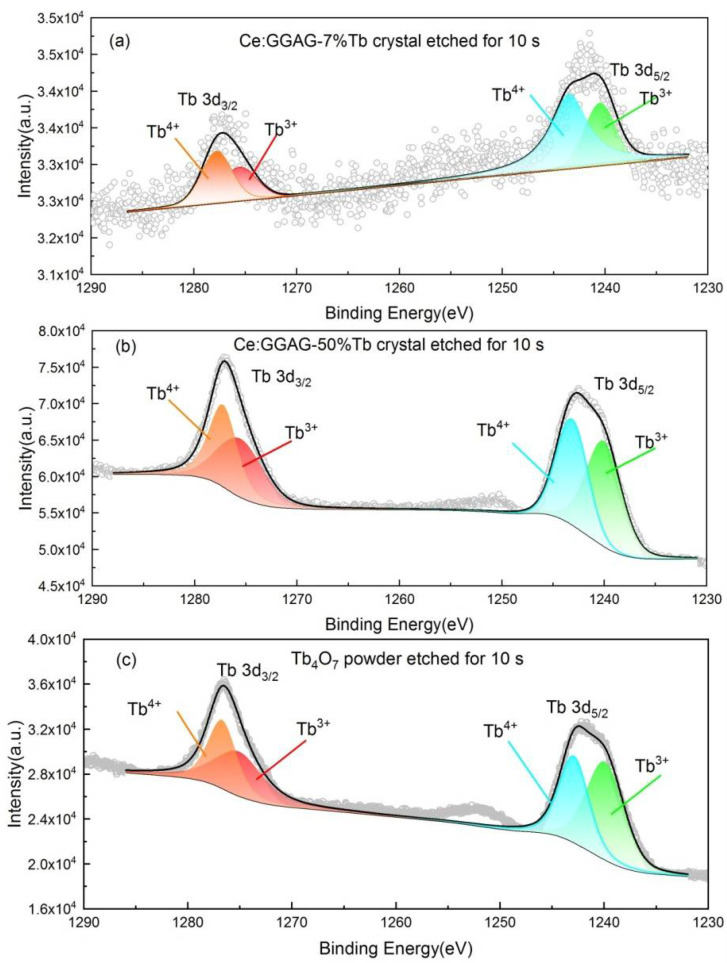
XPS spectra of the as grown (**a**) GGAG:Ce, 7% and (**b**) GGAG:Ce , 50% Tb^3+^ crystals using commercial (**c**) Tb_4_O_7_ powder as reference sample.

**Figure 4 materials-15-02044-f004:**
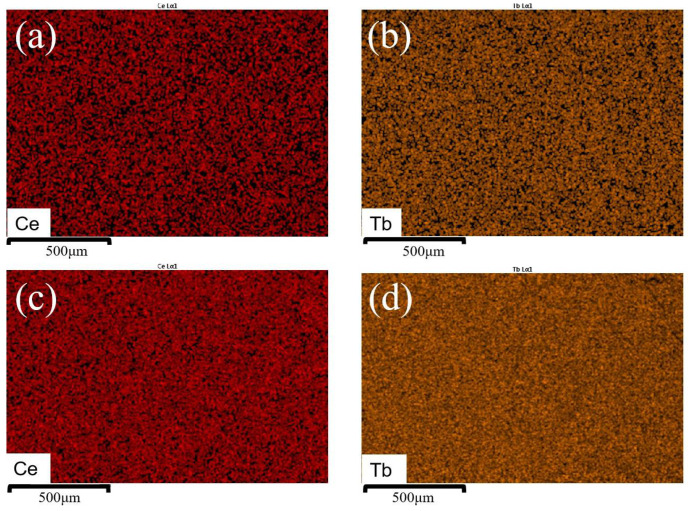
SEM-EDS mapping images for (**a**) Ce element (**b**) Tb element of GGAG:Ce, 7% Tb^3+^ crystal and (**c**) Ce element (**d**) Tb element of GGAG:Ce, 50% Tb^3+^ crystal.

**Figure 5 materials-15-02044-f005:**
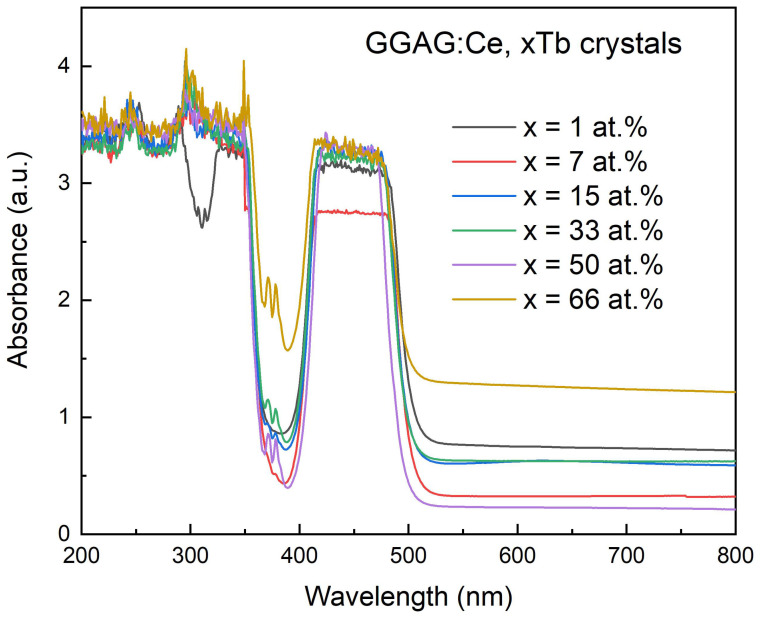
Absorptance spectra of the as grown GGAG:Ce, xTb (x = 0–66 at.%) crystals (after double face polished to1.0 mm thickness).

**Figure 6 materials-15-02044-f006:**
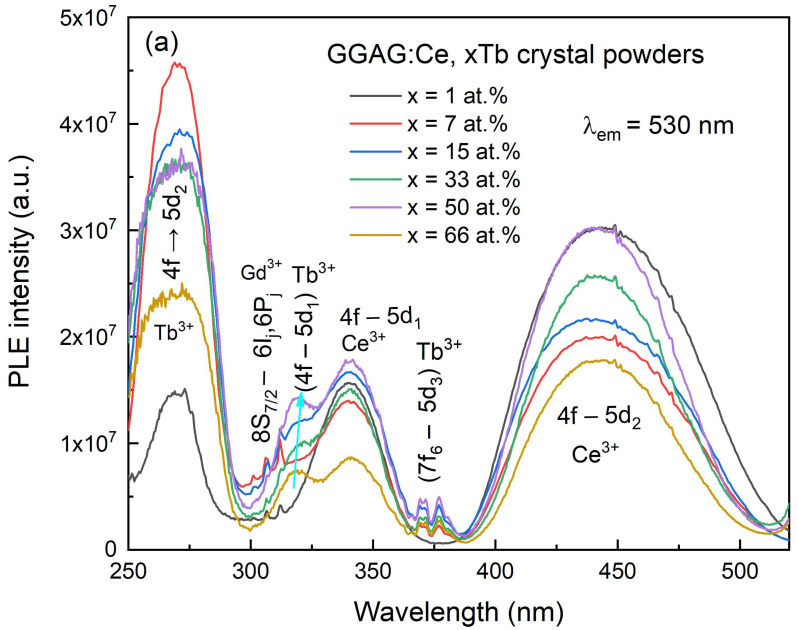
Photo-luminescence spectra of the as grown GGAG:Ce, xTb (x = 0–66 at.%) crystals; (**a**) PLE λ_em_ = 530 nm; (**b**) PL λ_ex_ = 270 nm; (**c**) PL λ_ex_ = 320 nm; (**d**) PL λ_ex_ = 450 nm; (**e**) Normalized PL spectra, λ_ex_ = 450 nm.

**Figure 7 materials-15-02044-f007:**
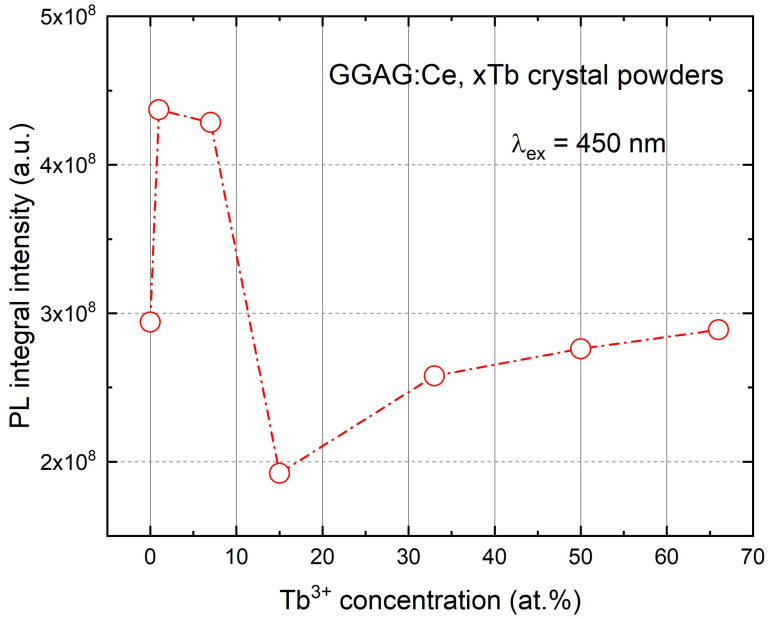
Photo-luminescence integral intensity of the as grown GGAG:Ce, xTb (x = 0–66 at.%) crystals, λ_ex_ = 450 nm.

**Figure 8 materials-15-02044-f008:**
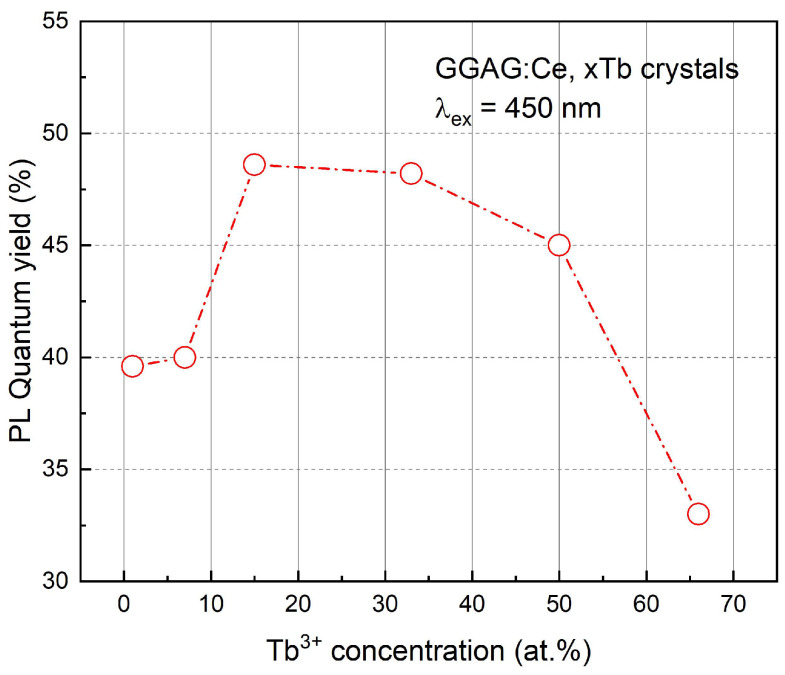
The PL QY of the as-grown GGAG:Ce, xTb (x = 0–66 at.%) crystals.

**Figure 9 materials-15-02044-f009:**
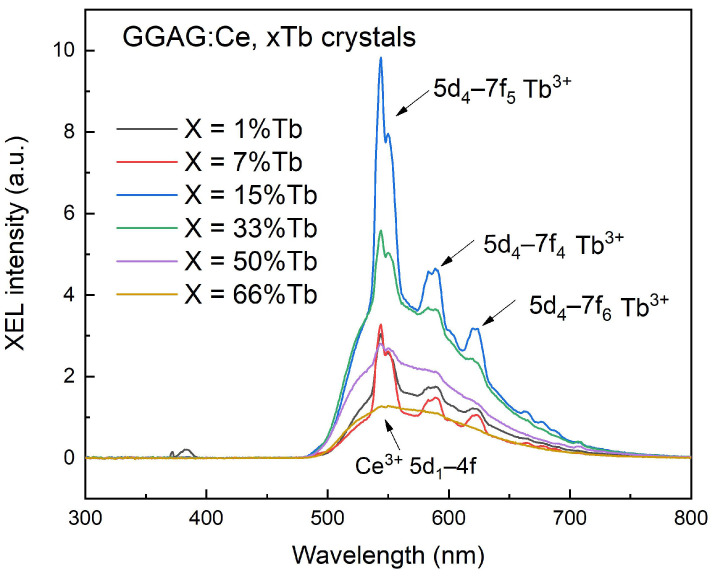
Radio-luminescence spectra of the as grown GGAG:Ce, xTb (x = 0–66 at.%) crystals, X-ray tube: 70 kV, 1.5 mA.

**Figure 10 materials-15-02044-f010:**
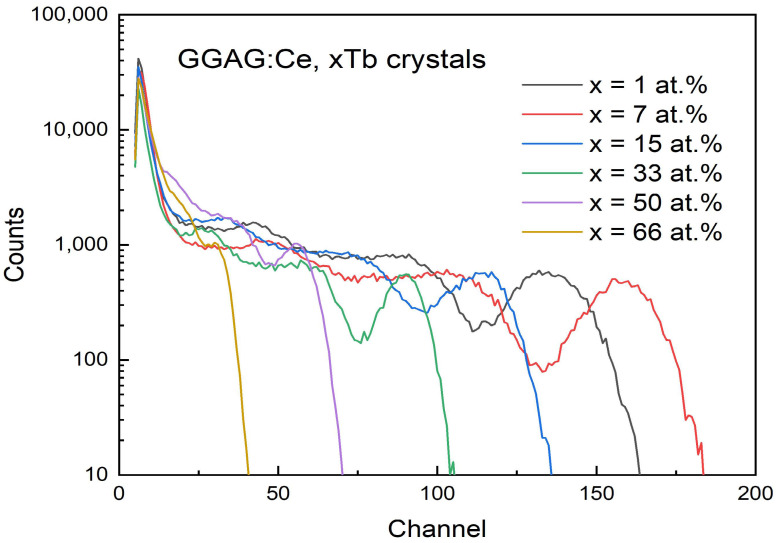
Pulse height spectra of the as grown GGAG:Ce, xTb (x = 0–66 at.%) crystals. The shaping time is 0.75 μs, under ^137^Cs (662 keV) gamma ray.

**Figure 11 materials-15-02044-f011:**
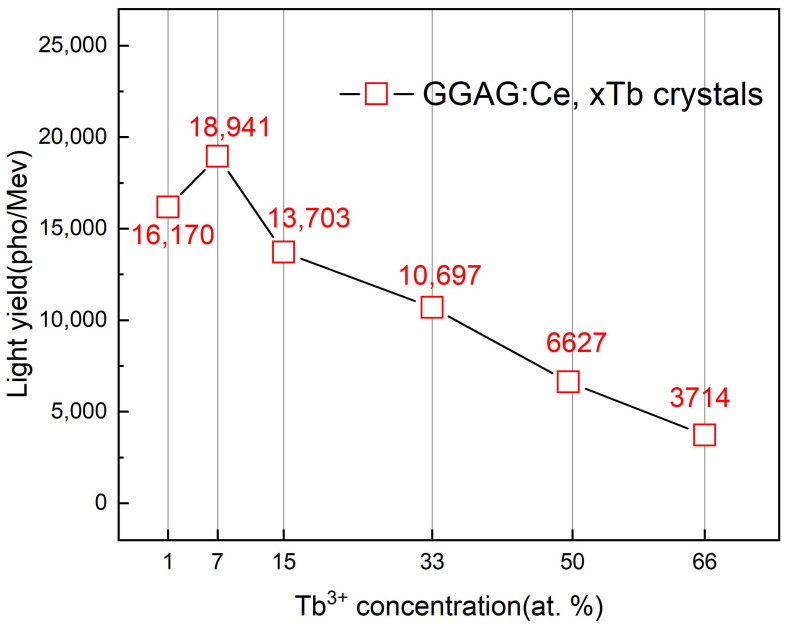
Calculated relative light yield of as grown GGAG:Ce, xTb (x = 0–66 at.%) crystals with respect to the reference GGAG:Ce crystal (LY 58,000 pho/MeV).

**Table 1 materials-15-02044-t001:** The Binding Energy (eV) and Tb^3+^ ratio in the GGAG:Ce, 7% Tb and GGAG:Ce, 50% Tb crystals by fitting the XPS spectra and comparing with high pure commercial Tb_4_O_7_ powder as reference.

Sample	Binding Energy (eV)				Radio
	Tb 3d_3/2_		Tb 3d_5/2_		Tb^3+^/ (Tb^3+^ + Tb^4+^)
	Tb^4+^	Tb^3+^	Tb^4+^	Tb^3+^	
GGAG:Ce, 7% Tb	1277.8	1275.6	1243.5	1240.5	
Integral area	2782	1968	6397	3205	36%
GGAG:Ce, 50% Tb	1277.3	1275.6	1243.2	1240.1	
Integral area	47,177	46,831	57,342	66,188	52%
Tb_4_O_7_ powder	1276.7	1275.1	1242.9	1239.9	
Integral area	24,442	26,652	31,212	41,573	55%

**Table 2 materials-15-02044-t002:** The SEM-EDS mapping analysis for chemical concentration of the Ce element, Tb element, Gd element, Ga element and Al element in the obtained GGAG:Ce, xTb crystals.

Nominal Chemical Composition of GGAG:Ce, xTb Crystals	Experimental Mol Value of Different Elements in Per Mol Compositions by SEM-EDS				
	Ce	Tb	Gd	Ga	Al
Ce_0.03_Tb_0.03_Gd_2.94_Ga_2_Al_3_O_12_	0.02	0.04	2.95	1.68	3.32
Ce_0.03_Tb_0.21_Gd_2.76_Ga_2_Al_3_O_12_	0.02	0.20	2.71	1.88	3.18
Ce_0.03_Tb_0.45_Gd_2.52_Ga_2_Al_3_O_12_	0.02	0.45	2.48	1.82	3.24
Ce_0.03_Tb_0.99_Gd_1.98_Ga_2_Al_3_O_12_	0.02	0.96	1.94	1.84	3.24
Ce_0.03_Tb_1.485_Gd_1.485_Ga_2_Al_3_O_12_	0.02	1.43	1.49	1.78	3.28
Ce_0.03_Tb_1.98_Gd_0.99_Ga_2_A_3_O_12_	0.02	1.97	0.98	1.70	3.33

## Data Availability

https://pan.baidu.com/s/1iTxKriBGW6mcevbDYSY83A?pwd=lfuj, accessed on 15 December 2012, Code:lfuj.

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
