# Peer review of "Wide Concentration Range of Tb3+ Doping Influence on Scintillation Properties of (Ce, Tb, Gd)3Ga2Al3O12 Crystals Grown by the Optical Floating Zone Method"

_materials, 2022, doi:10.3390/ma15062044_

Round 1

Reviewer 1 Report

REVIEWER COMMENTS

Comment 1: (Ce0.1TbxGd1-x)3Ga2Al3O12 (x = 0, 1%, 7%, 15%, 33%, 50% and 66%) Tb3+ ratios are taken. According to these rates, the Tb3+ concentration is 3 times higher. These ratios are valid for (Ce0.01)TbxGd3-xGa2Al3O12 formulation. It should be reviewed and seven compounds should be given. Also 1% Ce = 0.01Ce, not 0.1Ce.

Comment 2: XRD reflections should be added in Fig. 2, and figure caption should be rearranged by specifying the production methods.

Comment 3: Gd ion radius (0.938) Tb ion radius (0.924) are close to each other. In addition, in a synthesis made by atomic displacement, the secondary phase effect is expected not to be at the rate of 1%. In contrast, there is a fairly high secondary phase. On the other hand, the secondary phase decreased a high Tb3+ ratio. For example, the secondary phase Ä°ntensity occurring at 1% at% is lower than 50% at%. What is the reason for this?

Comment 4: RE3+ (RE=Gd, Tb, Ce) transitions should be shown in Fig. 4 (PL) and Fig. 6 (RL) spectra.

Comment 5: In Fig. 6, why is the RL (radioluminescence) peaking at 15 at% and peaking at 1% at PL (photoluminescence)?

Comment 6: The similarity rate (50%) should be reduced.

Reviewer 2 Report

Referee Report

“Wide concentration range of Tb doping influence on scintillation properties of Gd3Ga2Al3O12 crystals grown by the optical floating zone method”

by K. Tong Wu, Ling Wang, Yun Shi, Xintang Huang, Qian Zhang, Yifei Xiong, Hui Wang, Jinghong Fang, Jinqi Ni, Huan He, Chaoyue Wang, Zhenzhen Zhou, Qian Liu, Qin Li, Jianding Yu,  Oleg Shichalin, Evgeniy Papynov

submitted to Materials

This article is devoted to an undoubtedly relevant topic. The synthesis, structure and optical properties of scintillator Ce doped (TbxGd1-x)3Ga2Al3O12 crystals are discussed. To begin with, I want to note that the article contains the results of the study of luminescence and scintillation properties, which are very interesting. However, in general, the article makes a negative impression and I will not recommend it for publication in Materials in present form for several reasons.

Firstly, the article is written very sloppy and requires editing by the authors. There is no unification of terminology, including samples names. The title refers to Tb substituted crystals, and then it refers to Ce substituted. There are a lit of mistakes in text.

Secondly, the authors write that studies of the chemical composition have been made (p. 2, lines 80-84), but the results are not presented. That is, the composition is not confirmed.

Finally, I am not satisfied with the description of the results of the phase composition study. The authors claim that only 7 at. % Tb crystal reached a single phase of garnet while the rest single crystals (1%, 15, 33 50 and 66 at % of Tb) are mixed phases of garnet and perovskite. There is no explanation for the reasons why only 7 at.%, why not 1%? This is an obvious anomaly and requires analysis, as it can greatly affect the optical properties. This raises doubts about the quality of the synthesis process. Thus, it is possible that the samples could be confused. This is supported by the fact that in Figure 4, 7 and 8 a linear relationship will be built if samples with 1 at% and 7at% are interchanged.

Reviewer 3 Report

In this study, the authors prepared Al-based garnets with different Tb concentrations and investigated their scintillation properties. In general, the manuscript can be reconsidered for publication after a major revision. 

1) Calculate the % of garnet and perovskite phases for each composition. 
2) The chemical composition of the final samples should be verified by some quantitative method (XRF, AAS, ICP-MS/OES, etc.) because some elements can be lost during the ball milling process.
3) Provide XPS analysis - it is required to see the oxidation states of Ce and Tb in each sample. 
4) Uniform distribution of all elements should be verified by EDX ELEMENTAL mapping over some selected areas - to confirm the absence of dopants segregation. 
5) XRD patters - label all diffraction peaks 
6) The authors should compare their samples by QY, not by intensity - as the emission intensity depends on numerous factors.
7) Please comment on the reproducibility and of the samples and results, how many samples were tested per trial?      

Round 2

Reviewer 1 Report

The manuscript is acceptable after a general language check and spelling corrections (such as Figure 6b).

Author Response

Thank you very much for your constructive comments which help to not only improve the quality of our manuscript but also deepen our understand in this field. We have revised the manuscript according to the comments as following. All the revisions were marked in red color in the revised manuscript.

Reviewer 2 Report

I will not recommend it for publication in Materials.

Author Response

Thank you very much for your constructive comments which help to not only improve the quality of our manuscript but also deepen our understand in this field. We have revised the manuscript according to the comments as following. All the revisions were marked in red color in the revised manuscript. Please see the attachment for the detail reply.

Reviewer 3 Report

In general, the manuscript should be revised - some important aspects are missing!

2) Chemical composition, i.e. real Tb concentration must be revealed - PL properties are strongly depends on activator concentration! No one knows how much activator remain in the sample after ball milling process! 

6) As it was stated before - samples must be compared by QY, not intensity, many works now wrongly report only intensity improvement! 

7) One sample per batch is not a real science - authors must test at least three to ensure the reproducibility!  

Round 3

Reviewer 3 Report

No more comments